# Comparative Evaluation of Quality and Metabolite Profiles in *Meju* Using Starter Cultures of *Bacillus velezensis* and *Aspergillus oryzae*

**DOI:** 10.3390/foods11010068

**Published:** 2021-12-28

**Authors:** Na-Young Gil, Ye-Ji Jang, Hee-Min Gwon, Woo-Soo Jeong, Soo-Hwan Yeo, So-Young Kim

**Affiliations:** Fermented and Processed Food Science Division, Department of Agrofood Resources, National Institute of Agricultural Science, RDA, 166, Nongsaengmyeong-ro, Iseo-myeon, Wanju 55365, Korea; 01077510220@hanmail.net (N.-Y.G.); rowlsla94@korea.kr (Y.-J.J.); vitamin89@korea.kr (H.-M.G.); wjddntnek@korea.kr (W.-S.J.); yeobio@korea.kr (S.-H.Y.)

**Keywords:** *Aspergillus oryzae*, *Bacillus velezensis*, *Meju*, metabolites

## Abstract

The production of good *Meju* soybean paste primarily depends on the selection of raw materials and fermenting microorganisms, which together influence its characteristic metabolites, taste, and aroma. In this study, we analyzed the relationship between properties and metabolites in *Meju* samples fermented by *Aspergillus oryzae* alone or with *Bacillus velezensis*. We developed fast-stable processing techniques to obtain *Meju* from *A. oryzae* and *B. velezensis* using the inoculation method, thereby ensuring safety in the production of soybean paste. The amino-type nitrogen content increased from an initial 180–228 mg% to a final 226–776 mg% during fermentation and was higher in *Meju* inoculated separately with the fungi and bacteria (C group) than in *Meju* co-inoculated with both the starters concurrently (D group). The levels of metabolites such as glucose, myo-inositol, glycerol, and fatty acids (palmitic, stearic, oleic, and linoleic acids) in *Meju* fermented by *A. oryzae* with *B. velezensis* were higher than those in *Meju* fermented by *A. oryzae* alone. Fungal growth was affected by the inoculated bacteria, which often occurs during the fermentation of co-inoculated *Meju*.

## 1. Introduction

Soybeans are an important raw material commonly used in food manufacturing in Asian countries, including Korea, China, Indonesia, and Japan, owing to their high content of protein and oil (approximately 40% and 20% of their dry weight, respectively) [1]. Many studies have been conducted on the salubrious effects of soybeans and their products, such as soybean paste and sauce, particularly the anti-cancer effects and amelioration of cardiovascular and other chronic diseases, indicating that this is an important food ingredient [2,3].

*Meju*, an unsalted representataive of fermented soybean food in Korea, is used in the production of soybean paste (*Doenjang*), soy sauce (*Ganjang*), and red pepper paste (*Gochujang*). *Meju* is classified into two types, traditional and modified, according to the manufacturing method. Traditional *Meju* is naturally fermented by mold or bacteria while the steamed beans dry, whereas modified *Meju* is koji made by inoculating the microbial starters into raw starch materials, such as rice, wheat, or barley [4]. Various fungi (including the genera *Aspergillus, Mucor,* and *Rhizopus*) and bacteria (including the genera *Bacillus* and *Staphylococcus*) have been found in *Meju* during fermentation [5,6]. Owing to their capacity to produce various enzymes, these microorganisms can degrade proteins and carbohydrates into small molecule compounds, such as amino acids and fatty acids [7]. Of these microorganisms, *Aspergillus oryzae* has been widely used in the food industry as a starter to manufacturing modified *Meju*, but many types of bacilli and fungi have been found in *Meju*. However, biogenic amines, such as tyramine, histamine, and putrescine, or biohazards, such as *Bacillus cereus* or *Aspergillus flavus*, can also occasionally occur [8]. Therefore, many studies have attempted to use a bacilli starter to control these hazards in the production of traditional fermented soybean products. Moreoever, a single starter, such as bacilli, that reduces the relevant harmful factors for these products to solve these problems has been developed. However, a bacilli starter inhibits useful fungi, such as *A. oryzae*. Accordingly, a solution to co-grow both bacilli and mold starters should be developed.

*Bacillus* sp. produces antifungal and antimicrobial agents, but research using it as a starter for fermented soy products is lacking [9]. The Ministry of Food and Drug Safety limits edible bacterial to only six species of *Bacillus*: *B. amyloliquefaciens*, *B. velezensis*, *B. coagulans*, *B. natto*, *B. polyfermenticus*, and *B. subtilis* [10]. *B. velezensis* was proposed as a subspecies strain of *B. amyloliquefaciens* subsp. *plantarum* but has recently been recognized by physical analysis as a synonym for *B. methylotrophicus* [11,12]. *B. velezensis* has been reported to have salt tolerance and high enzymatic activity [13]. Studies on *B. velezensis* have isolated and characterized strains with antifungal activity from Korean traditional soy sauce for application as starters [14,15].

The aim of study was to compare the fermentative changes associated with using *B. velezensis* with *A. oryzae* as starters in *Meju* fermentation and to investigate the changes in quality characteristics, metabolites, and microbial community using gas chromatography–mass spectrometry (GC-MS) and next-generation sequencing analysis. Furthermore, *B. velezensis* was evaluated by establishing a manufacturing process that considers antagonism with *A. oryzae*.

## 2. Materials and Methods

### 2.1. Manufacturing of the Modified Meju

In this study, we prepared *Meju* by using the modified method with bacilli and fungi starters to ferment soybeans. Based on a previous study, *B. amyloliquefaciens* NY12-2 (reclassified as *B. velezensis* NY12-2, Accession: NZ_CP033576), which is used to ferment *Meju*, was selected as a bacterium that inhibits *B. cereus* [16]. *A. oryzae* KCTC 46471 was purchased from the Korean Collection for Type Cultures (KCTC, Daejeon, Korea). The experimental groups were divided into four *Meju*: control without a starter (Group (A)), *A. oryzae* single inoculation (Group (B)), *B. velezensis* and *A. oryzae* individual inoculation in each other chamber Group (C), and *B. velezensis* and *A. oryzae* co-inoculation in a chamber Group (D).

After cleaning and soaking 10 kg of soybeans (Daewon cultivar (Andong, Korea)) in water for 15 h, the soybeans were steamed at 121 °C for 60 min. The cooled steamed beans were inoculated with *B. velezensis* NY12-2 (1 × 10^8^ CFU/mL in saline, 1%) and *A. oryzae* KCTC 46471 (1 × 10^7^ spores/mL in saline, 1%) and then incubated at 30 °C for 1 day.

### 2.2. Physicochemical Properties Analysis

The moisture content of *Meju* was analyzed at 105 °C by drying according to the Association of Official Analytical Chemists method [17]. Two grams of each sample were dried in a 105 °C dry oven (MOV-112, Sanyo Co., Ltd., Osaka, Japan) until a constant weight was reached, which was expressed as a percentage.

For preparation for pH measurement, 50 g of *Meju* samples were diluted five times with 200 mL of distilled water, homogenized using a PT-MR 2100 homogenizer (Kinematica, Lucerne, Switzerland) for 1 min, and filtered through Whatman No. 2 filter paper. pH was measured using an Orion star a211 pH meter (Thermo Fisher Scientific, Beverly, MA, USA). The acidity was calculated by adding 0.5% phenolphthalein as an indicator to 10 mL of the sample solution and titrating with 0.1 N NaOH until it became reddish. The amino-type nitrogen content was measured via the formalin titration method. The pH of the sample solution was adjusted to 8.4 by adding 0.1 N NaOH; then, 10 mL of neutral formalin was added to the solution, which was then titrated with NaOH (0.1 N) to reach pH 8.3–8.4.

### 2.3. Total Viable Counts and MiSeq Metagenomic Sequencing

To count the total viable bacteria and fungi in each *Meju* sample, 1 g of sample was mixed in 9 mL of physiological saline (0.85% (*w*/*v*) NaCl) to accomplish a ten-fold dilution and homogenized using a Vortex-Genie 2 (Scientific Industries, Inc., Bohemia, NY, USA). The homogenate was serially diluted with saline solution. Diluted samples (0.05 mL) were inoculated on tryptic soy agar or potato dextrose agar plates and incubated for 24 or 48 h at 30 °C, respectively. Total counts were calculated as typical colonies identified on each agar plate, considering dilutions presumed to be bacilli (with dehydrated big colony) or mold (with mycelium; for comparison, yeast forms a white opaque colony).

Metagenome sequencing was performed to identify microbial communities in different *Meju* environments. DNA was extracted using a PowerSoil® DNA Isolation Kit (Cat. No. 12888, MO BIO, Carlsbad, CA, USA) according to the manufacturer’s protocol. Each sequenced sample was prepared according to the Illumina Sequencing Library protocols. DNA quantification and quality were analyzed using Quant-iT™ PicoGreen and Nanodrop ND-1000 (Thermo Fisher Scientific, Beverly, MA, USA). The 16S rRNA genes were amplified using V3–V4 region (forward: 5′-TCGTCGGCAGCGTCAGATGTGTATAAGAGACAGCCTACGGGNGGCWGCAG-3′; reverse: 5′-GTCTCGTGGGCTCGGAGATGTGTATAAGAGACAGGACTACHVGGGTATCTAATCC-3′) and ITS3–ITS4 (forward: 5′-TCGTCGGCAGCGTCAGATGTGTATAAGAGACAGGCATCGATGAAGAACGCAGC-3′; reverse: 5′-GTCTCGTGGGCTCGGAGATGTGTATAAGAGACAGTCCTCCGCTTATTGATATGC-3′) primers. Input gDNA was amplified with 16S rRNA V3–V4 or 5.8–28S rRNA ITS3–ITS4 primer sets, and multiplexing indices and Illumina sequencing adapters were added by subsequent limited-cycle amplification. The final products were normalized and pooled using PicoGreen, and the library sizes were verified using the TapeStation DNA D1000 screen tape assay (Agilent, Santa Clara, CA, USA). The libraries were then sequenced using MiSeq^™^ (Illumina, San Diego, CA, USA) by Macrogen (Seoul, Korea).

### 2.4. Microbial Community Metagenome Using PICRUSt Based on 16S rRNA Sequencing Data

PICRUSt analysis was used to predict the microbial community metagenome using the Greengene database based on taxonomic abundance, as detailed by Langille et al. [18].

All predicted gene families were classified using the Kyoto Encyclopedia of Genes and Genomes (KEGG) pathway database (https://www.genome.jp/kegg/kegg2.html, accessed on 19 May 2020). The functions of genes and their relative abundances in the predicted metagenome were compared among the *Meju* samples and visualized in a heatmap using the omics function in XLStat (XLStat).

### 2.5. Volatile Analysis by GC-MS

The volatile compounds were measured by adding 1 g of sample, 1 g of NaCl, and 4 mL of distilled water (including internal standard) into a 20 mL glass vial, heating at 40 °C, stirring for 15 min, and adsorbing for 10 min with solid-phase microextraction (SPME). The volatile compounds were analyzed using a GC-2010 Plus gas chromatograph (Shimadzu, Kyoto, Japan) equipped with a DB-WAX column (30 mm × 0.25 mm i.d. × 0.25 μm film; J & W Scientific, Folsom, CA, USA). The flow rate of the helium carrier gas was maintained at 1 mL/min, and the injector temperature was set to 250 °C. The initial oven temperature was maintained at 40 °C for 3 min, raised to 90 °C at 5 °C/min, raised again to 230 °C at 19 °C/min, and then maintained for 5 min. The mass spectra were analyzed using a GC-MS-TQ 8030 (Shimadzu) and monitored in the Q3 scan mode. The temperatures of the ion interface and source were set to 280 °C and 230 °C, respectively. A detector voltage of 0.1 kV was used.

### 2.6. Metabolite Analysis

For metabolite analysis, *Meju* at a concentration of 0.02 g/500 μL in 80% methanol was extracted using a blender. The extract was centrifuged for 10 min at 16,127× *g* and 25 °C, and then 10 μL of the obtained supernatant was dried using a speed vac concentrator (Labconco Co., Kansas City, MO, USA). Derivatization was performed at 37 °C for 90 min after adding 70 μL hydroxymethoxy amine. Subsequently, 70 μL of *N*,*O*-*bis*(trimethylsilyl)trifluoroacetamide was added, and the mixture was incubated to 70 °C for 30 min. The derivatized reactant was centrifuged for 10 min at 16,127× *g* to obtain the supernatant. Relative quantification of the metabolites was performed using GC-MS.

Metabolites of the *Meju* samples were analyzed using a GC-2010 Plus (Shimadzu, Tokyo, Japan) equipped with a DB-WAX column (30 m × 0.25 mm i.d. × 0.25 μm; J&W Scientific, Santa Clara, CA, USA). The flow rate of the helium carrier gas was maintained at 1 mL/min, and the injector temperature was set at 200 °C. The oven temperature was held at 80 °C for 2 min and then raised at 10 °C/min to 320 °C, where it was maintained for 6 min. MS analysis was performed under the same conditions.

MS data analysis was conducted using Pro software (Spectralworks Ltd., Runcorn, UK). Metabolites were calculated using n-alkanes and confirmed by comparing the mass spectra and retention indices with the Wiley and National Institute of Standards and Technology mass spectral databases. The resulting data were input to SIMCA-P+ version 12.0 (Umetrics, Umea, Sweden) for collection, alignment, and normalization. Metabolite profiles were displayed as heatmaps using XLStat (XLStat, New York, NY, USA).

### 2.7. Statistical Analysis

All experiments were performed in triplicate, and the analyzed data are presented as means ± standard deviation (SD). MS data were visualized by principal component analysis (PCA) and partial least squares discriminant analysis (PLS-DA). *p*-values were determined using SAS software (SAS Institute, Cary, NC, USA), and those below the threshold of *p* = 0.05 were considered statistically significant. PCA and heatmap analysis were performed using XLStat for Microsoft Excel (Microsoft Corp., Redmond, WA, USA). Significant differences of all data, including physicochemical components, viable microbial counts, and metabolites, were tested by one-way analysis of variance (ANOVA) with Duncan’s multiple range test using SPSS version 17.0 (SPSS Inc., Chicago, IL, USA).

## 3. Results and Discussion

In our study, we aimed to develop a fast-stable processing technique to obtain *Meju* from *A. oryzae* and *B. velezensis* by the inoculation method.

### 3.1. Physicochemical Properties of Modified Meju

We fermented *Meju* for 24 h at 30 °C using starters of *A. oryzae* with or without *B. velezensis*. The qualitative properties and viable microbial counts are presented in Table 1. The initial moisture and pH of all *Meju* samples were 58.18–60.12% and 6.46–6.62, respectively, which decreased to 57.66%–59.96% and 6.28–6.57, respectively, after 1 day of fermentation. There was no significant difference in the moisture content among the *Meju* samples. Group B had the lowest pH (6.28) after 24 h, but no notable difference was observed between the experimental groups and the fermentation period.

The acidity (0.95%) in sample C dramatically increased by the end of the fermentation process. The acidity of sample A did not change during the fermentation period, but that of sample B increased significantly more than that of sample D as the fermentation progressed (*p* < 0.05).

Here, we doubted the survival of *A. oryzae* in sample D. There are no published reports of *B. velezensis* inhibiting *A. oryzae*, which is non-pathogenic. However, *Bacillus* sp. has antifungal effects against molds, such as *Aspergillus niger* and *Rhizopus* spp. [19]. Furthermore, Choi et al. reported differences in fermentative activity and acid production depending on the fungal strain used [20]. In their study, *A. oryzae* and *Rhizopus oligosporus* showed titratable activity at 1.35–1.40% and 1.60–1.75%, respectively, which was higher than that in commercial *Meju* without fungi (0.88%).

In addition, the content of amino-type nitrogen is the key component in soybean fermentation, and the content can increase from an initial 180–228 mg% to a final 226–776 mg% during fermentation. Samples B and C had relatively high final content (776 mg%), whereas that of sample D did not increase during the same period. The high amino-type nitrogen content in fungi–bacteria-fermented *Meju* can be correlated to the viability of *A. oryzae* by the antifungal effect of *B. velezensis*. *B. velezensis* inhibits the growth of pathogenic fungi, such as *A. flavus*, *Fusarium graminearum*, and *A. oryzae,* and reduces their ability to produce the mycotoxins aflatoxin and ochratoxin [21].

### 3.2. Effect of Starter Inoculation Method on Changes in Bacterial and Fungal Communities

The total viable bacterial cell count in all experimental groups increased after fermentation, with Group C having the highest count (9.40 ± 0.02 cfu/g; Table 1). Mold was not found even after fermentation for 24 h in Group A, but the mold count increased in Groups B and C. In the case of Group D, the total decreased as fermentation progressed. *B. cereus* was not detected in any experimental group.

In this study, the diversity of bacteria and fungi did not vary during the improved *Meju* fermentation using starters. The sequencing reads for the bacteria and fungi were classified at the genus and species levels to compare changes in the microbial communities among the four groups after 1 day of fermentation. The evaluation of the bacterial community at the genus level indicated that the genera *Bacillus* and *Aerosakkonema* predominated; the relative abundances ratio in all samples was 3.6–94.0:15.5–86.3, respectively (Figure 1a). At the species level, bacterial community analysis showed that the relative abundances ratio of *B. velezensis* and *Aerosakkonema funiforme* before and after fermentation was notably different depending on the method used to inoculate starters. In Group B, there was no significant difference in bacterial composition before and after fermentation. *A. funiforme* initially predominated (86.3–86.8%), but after 24 h, *B. velezensis* was detected at a level of 4% without starter inoculation. In contrast, *B. velezensis* in samples C and D increased from 60.7% to 94.0% and from 49.0% to 86.3%, respectively. In addition, *A. funiforme* in samples C and D decreased from 37.5% to 5.9% and from 43.4% to 13.5%, respectively. Immediately after inoculation, other bacteria found in samples C and D showed the relative abundances ratio of 1.5–7.6%. After 1 day, the relative abundance ratio of the other bacteria decreased to 0.1–0.2% owing to the growth of bacillus spp (except for Group B, which showed a low relative abundance ratio of bacilli). *A. funiforme,* which is included in the phylum Cyanobacteria, was first reported to be isolated from mesotrophic reservoirs by Thu et al. [22]. To the best of our knowledge, this is the first report of *A. funiforme* being predominant in *Meju*.

Phylum-level analysis of the fungal community revealed that unassigned phyla and *Aspergillus* were the major eukaryotes in the *Meju* samples (Figure 1b), which agrees with the results of a previous study [23]. In all inoculated *Meju* samples, *Aspergillus* was predominant with a relative abundance of 24.9–96.8%, and the relative abundances of unassigned eukaryotes were also high at 3.2–73.1%. However, their relative abundances before and after fermentation varied depending on the starter inoculation method. These results indicate that the growth of *A. oryzae* is closely related to an increase in fungal abundance. However, the relative abundance ratio of *Aspergillus* in sample B was higher than that in sample D because of the antifungal effect of *B. velezensis* [21]. These results suggest that the growth of fungi in the presence of the co-inoculated bacteria often reduces during fermentation. When *Meju* is manufactured using combination starters, we should consider their antagonism effect and suitable inoculation method.

### 3.3. Volatile Components

The flavor characteristics of the fermented *Meju* samples were analyzed using GC-MS and headspace SPME. The total of 35 volatile compounds identified in all *Meju* samples included 10 esters, 13 alcohols, 1 acid, 2 aldehydes, 6 ketones, 2 pyrazines, and 1 other species. The main flavor volatiles, ethyl acetate, methylene chloride, and ethanol, were detected at a high abundance at 0 h of fermentation. The content of ethyl acetate and ethanol increased, whereas that of methylene chloride decreased after 24 h of fermentation. Ethanol is produced by the reduction in carbonyl compounds of aldehydes, which correspond to soybean protein or fat metabolites, but most have been reported as metabolites converting glucose to ethanol during soybean fermentation.

The 3-methyl-1-butanol content was relatively high in Groups B and C. This component produces soybean odor as a degradation product of leucine, which decreases upon heating. 2,3-Butanediol was produced during fermentation and was an intermediate of 3-hydroxy-2-butanone (acetoin) produced by polymerization with the thiamine pyrophosphate catalyst of ethanol. The proportions of other flavor components were generally less than 1%, with no significant difference among the *Meju* samples. This is because the flavor component of *Meju* is produced by the Maillard reaction during the fermentation of soy protein, fat, and carbohydrates and various enzymatic metabolic reactions by microorganisms and enzymes used in *Meju* production [24].

Ethyl acetate, ethanol, propyl acetate, ethyl hexanoate, ethyl 2-hydroxypropanoate, and 1-octen-3-ol are found in alcoholic beverages, which are fermented by molds and yeasts. Twelve compounds (2-propanone, methyl acetate, 2-pentanone, isobutyl acetate, 2-methyl-1-propanol, ethyl 2-methylbutyrate, 3-methyl-1-butyl acetate, 5-methyl-2-hexanone, 2-heptanone, 1-pentanol, 3-hydroxy-2-butanone, and 1-hexanol) have sweetish, fruity, banana, or buttery odors. In contrast, 2-methylpropanoic acid, 3-methyl-1-butanol, and benzaldehyde produce odors such as rancid butter, pungent, and bitter almond [25].

The content of 2-pentanone, isobutyl acetate, 3-hydroxy-2-butanone, 2-heptanone, 2,5-dimethylpyrazine, and benzaldehyde simultaneously increased after fermentation by *B. velezensis* (Figure 2). The odors of these compounds are sweet, such as banana, creamy, and almond. In contrast, ethyl acetate, ethanol, and 1-octen-3-ol were detected in *Meju* with high *A. oryzae* content. In particular, the content of 1-octen-3-ol, which has a mushroom odor, decreased in Groups C and D after fermentation with *Bacillus* starter. In addition, the content of 1-hexanol and 2-ethylhexanol, which have fresh grass and floral odors, was high in Group A and notably reduced in Groups C and D. Seo et al. reported that the contents of ethyl acetate in soybean koji fermented by *Bacillus amyloliquefaciens* or *A. oryzae* are 1.60 ± 0.11 and 2.49 ± 0.13 mg/kg, respectively. Moreover, koji fermented by *B. amyloliquefaciens* has a higher 2-methylpropanoic acid content (2.81 ± 0.15) than the koji fermented by *A. oryzae* (1.24 ± 0.04). In particular, ethyl 2-methylbutyrate (butanoate) is only detected in koji fermented with *B. amyloliquefaciens* (1.67 ± 0.07) [26]. Our results suggested that the *Meju* co-inoculated the Bacillus and fungi demonstrated improved flavor, whereas the one inoculated with only fungi did not.

Pent-1-en-3-ol, 4-ethenylpheol, 2-pheylethanol, and phenol were detected in samples fermented by *B**. velezensis*, whereas 2-methylpropanal and 2-phenylbut-2-enal were detected in those fermented by *A**. oryzae*. Moreover, benzaldehyde, 2-pentylfuran, hexanal, and 2-phenylacetaldehyde were detected in both.

### 3.4. Comparison of Metabolite Profiles

To evaluate the differences in primary metabolites among *Meju* samples, multi-factor analysis (MFA) and PLS-DA were performed using GC-TOF-MS or metagenome data in Figure 3. The MFA score plot showed 50.30% and 26.26% variances for F1 and F2, respectively. After fermentation, Groups B and C moved into quadrant 1, whereas Groups A and D moved into quadrants 3 and 4, respectively; each sample, fermented for one day, was separated in the MFA score plot, indicating that the *Meju* samples produced the metabolites, including fatty acids, free sugars, and sugar alcohols, owing to the inoculated starters. The quality parameters of the PLS-DA model were as follows: R2X = 0.796, R2Y = 0.285, and Q2 = 0.137 for bacteria and fermentation period metabolites. The values of the cross-validation (R2 intercept < 0.2 and Q2 intercept < −0.1) indicated the statistical acceptability of the PLS-DA models for bacterial and fungal metabolomic analysis, although the *p*-values in the PLS-DA score plots were higher than 0.05. Taken together, these results suggest that the metabolites of *Meju* during fermentation differ remarkably depending on the starter and inoculation method.

### 3.5. Heatmaps of Identified Metabolites

Among the metabolites found by GC-MS, the statistical results of the normalized metabolites showed the metabolites changed significantly during *Meju* fermentation. Thirty-three *Meju* metabolites were identified (data not shown). In particular, metabolites, such as organic acids (e.g., lactic acid, malic acid, 4-aminobutyric acid, and ketoglutaric acid), sugars (e.g., sucrose), and amino acids (e.g., alanine, oxoproline, asparagine, and tryptophan), with variable importance in projection values > 1.0 were identified as associated with energy metabolism. *A. oryzae* contributed more metabolites than *B. velezensis*.

Changes in the levels of metabolites identified in the four *Meju* groups were visualized according to metabolic pathways based on heatmaps (Figure 4). In Group B, the levels of metabolites, such as fructose, sorbitol, citric acid, and malic acid, were significantly higher than those in Groups C and D.

*Meju* in Group C contained metabolites of glucose, myo-inositol, glycerol, and fatty acids (palmitic, stearic, oleic, and linoleic acids), which were more highly detected in this other group than in the other groups. Of these metabolites, malonic acid and lactic acid were produced by *Bacillus*, supporting findings described by Seo et al. [26]. According to Kang, the tryptophan, lyxose, succinic acid, α-ketoglutaric acid, and 4-aminobutyric acid (GABA) content in soybeans (16.93 + 0.10) is higher than that in *Meju* and *Doenjang* (9.30 mg/100 g and 7.53 mg/100 g (dry basis), respectively) [27].

A correlation analysis conducted by Lee et al. showed that *B. velezensis* is linked to the production of fatty acids [28]. These results follow those of our study. The composition of Group C was 94% *B. velezensis* (Figure 1), and the production of fatty acids, such as palmitic acid, oleic acid, and linoleic acid, was higher than that in the other groups (Figure 3).

### 3.6. Predictive Functional Genes in Fungal and Bacterial Inoculants of Meju

The PICRUSt analytical tool is used in metagenomics and allows for inference of the functional profile of a microbial community based on a marker gene survey among samples [24]. In this study, to predict the genetic amino acid pathway from microorganisms, the *Meju* samples produced using *A. oryzae* inoculation with or without *B. velezensis* were aggregated into two clusters: pre- and post-fermentation (Figure 5).

For the initial fermentative samples, red clusters were related to the metabolism of alanine, aspartate, glutamate, cysteine, methionine, histidine, glycine, serine, threonine, and tryptophan and the degradation of lysine, valine, leucine, isoleucine, and tryptophan. Blue clusters were related to the metabolism of phenylalanine and tyrosine and biosynthesis of valine, leucine, isoleucine, phenylalanine, tyrosine, lysine, and amino acid-related enzymes for pre-fermentation. The difference between the clusters indicates there are more degradation function genes grouped in the red cluster and more biosynthesis function genes grouped in the blue cluster. The samples showed changes in their functional profiles at 24–48 h of fermentation compared with those at 0 h. Sample C showed a distinct amino acid profile among the starter-inoculated groups; at 0 h, it was classified into the same cluster as samples B and D at 24 h were, demonstrating that the fermentation rate of sample C was faster than the other samples. After 24 h, sample C showed metabolism of lysine, which is essential for nutrition, which moved it to the blue cluster. These results indicate that the inoculation method for sample C can predict the amino acid pathway from microbial communities. We propose that differences in inoculation determine the quality characteristics of *Meju*.

Microbial communities in fermented foods have been reported to have sensory characteristics such as those of Korean soybean paste. The amino acids degraded by bacteria are crucial factors that can determine the taste quality of soybean paste. Amino acids are responsible for sweet (lysine, alanine, glycine, serine, and threonine), umami (glutamate and aspartate), and bitter (isoleucine and phenylalanine) tastes [29,30,31,32]. Furthermore, this study showed that bacterial communities in *Meju* are responsible for the production of amino acids. Therefore, we focused on the function of bacteria in the predicted amino acid metabolism in *Meju*.

## 4. Conclusions

*Meju* is used in the production of soybean paste (Doenjang), soy sauce (Ganjang), and red pepper paste (Gochujang). The genera *Aspergillus* and *Bacillus* found in *Meju* during fermentation to influence its characteristic metabolites, taste, and aroma. However, *B. velezensis*, which is the most predominantly detected species in Korean soybean sauces, affects the growth of useful *A. oryzae*. Therefore, we investigated the differences in properties and metabolites in *Meju* samples fermented by *Aspergillus oryzae* alone or by *Bacillus velezensis*. We demonstrated that fermented *Meju*, with an amino-type nitrogen content of 776 mg%, can be manufactured using the fungi and bacteria separately (C group) over 1 day. Notably, the *Meju* produced from *A. oryzae* and *B. velezensis* using the inoculation method showed favorable metabolites, such as glucose, myo-inositol, glycerol, and fatty acids (palmitic, stearic, oleic, and linoleic acids), compared with *Meju* fermented by *A. oryzae* alone.

Consequently, we propose that the inoculation method using both fungal and bacterial starters can be effectively used in manufacturing *Meju* and contribute to fungal survival and fermentation.

## Figures and Tables

**Figure 1 foods-11-00068-f001:**
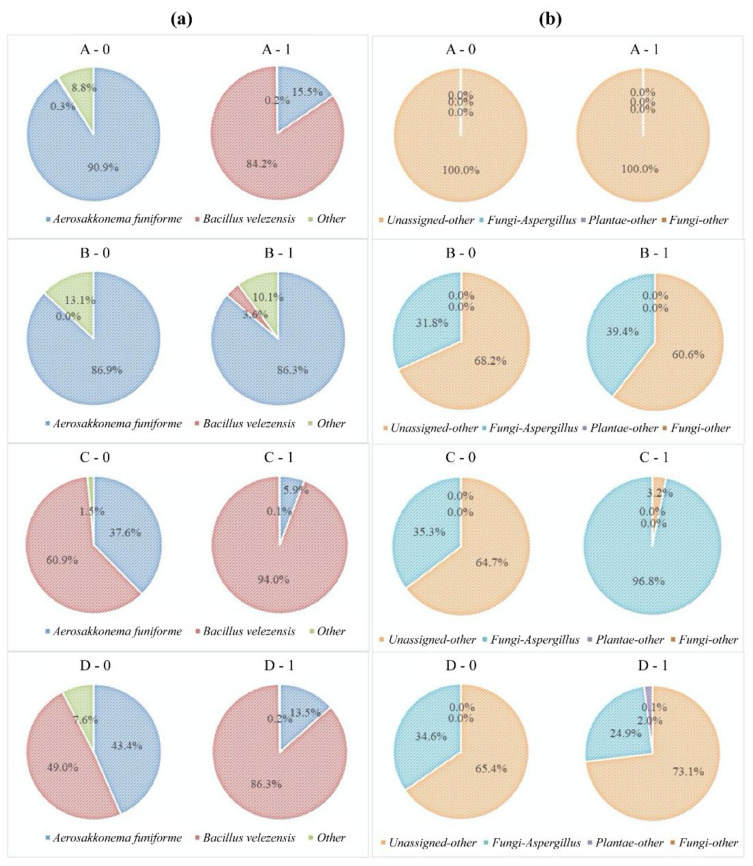
Taxonomic classification at the species level of the (**a**) bacterial 16S rRNA gene and (**b**) fungal ITS gene reveals changes in bacterial and fungal communities during one-day fermentation. The experimental groups were inoculated as follows: A, control without starter; B, *Aspergillus oryzae* single inoculation group; C, *Bacillus velezensis* and *A. oryzae* individual inoculation group; and D, *B. velezensis* and *A. oryzae* co-inoculation group.

**Figure 2 foods-11-00068-f002:**
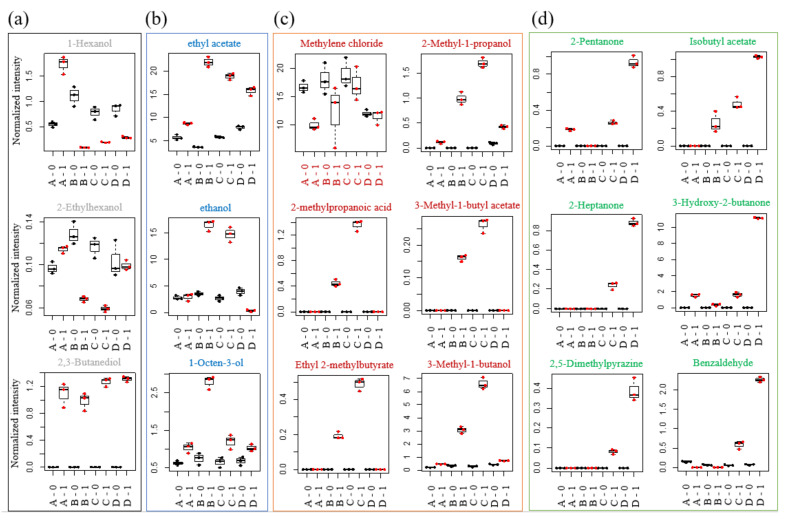
Boxplots derived from GC-MS data from *Meju* samples reveal metabolic changes during fermentation with different starters. (**a**) Volatiles increased in Group A as control without starter, (**b**) volatiles increased in Group B inoculated with *A. oryzae* KCTC 46471, (**c**) volatiles increased in Group C inoculated individually with *B. velezensis* and *A. oryzae*, and (**d**) volatiles increased in Group D co-inoculated with *B. velezensis* and *A. oryzae* between pre- and post-fermentation for 0–1 d.

**Figure 3 foods-11-00068-f003:**
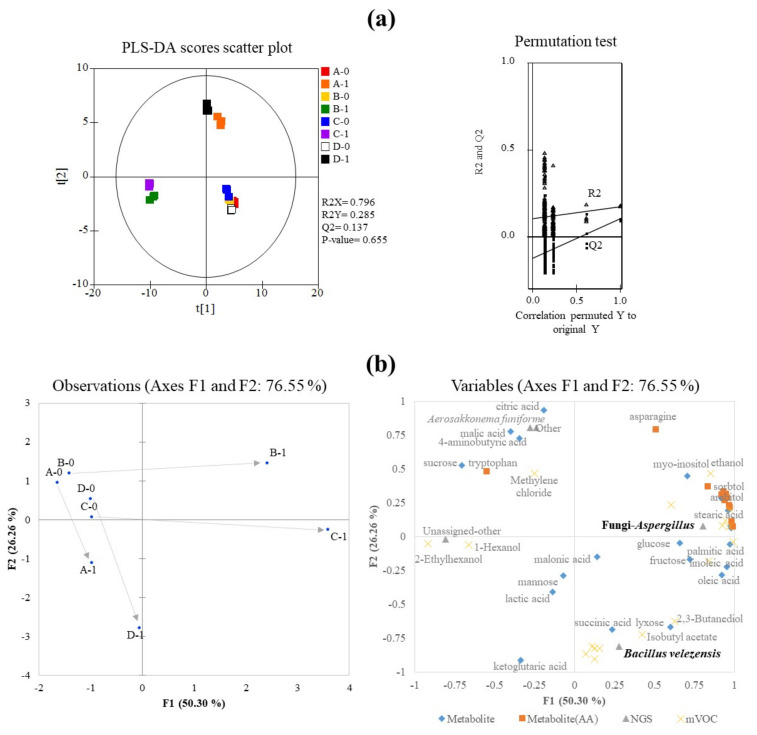
PLS-DA (**a**) was performed using GC-TOF-MS data and multi-factor analysis, (**b**) score plot from the set of GC-MS and NGS data from *Meju* samples reveal metabolic changes during fermentation among samples using different starters. Here, the datasets for raw substrates are indicated: ◆, metabolites dataset except for amino acids; ■, metabolites including only amino acids; ✕, microbial volatile datasets; ▲, metagenome datasets based on 16S rRNA and ITS sequences.

**Figure 4 foods-11-00068-f004:**
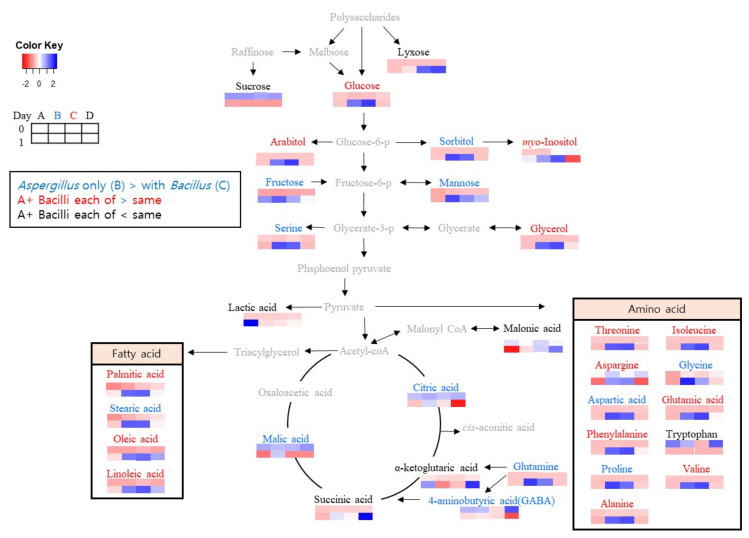
Scheme of a metabolic pathway based on the heatmap generated from GC-MS data from *Meju* samples reveals metabolic changes during fermentation for 0–24 h among samples using different starters. Metabolite compounds described as color of compound name: Blue, higher levels in Group B inoculated only *A. oryzae* than in Group C individually inoculated with *A. oryzae* and *B. velezensis*; Red, higher levels in Group C than in Group D co-inoculated simultaneously with *A. oryzae* and *B. velezensis*; Black, higher levels in Group D than in Group C.

**Figure 5 foods-11-00068-f005:**
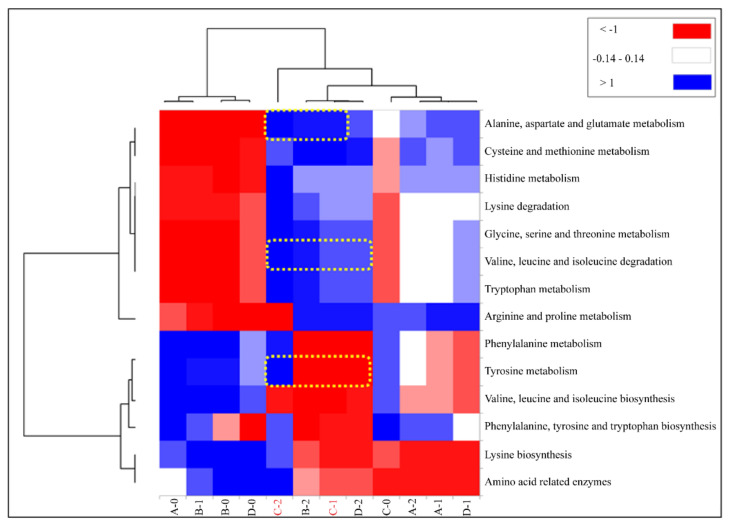
PICRUSt analysis of the operational taxonomic unit data of the metagenome reveals metabolic changes in fermented *Meju* using different starters. The experimental groups were inoculated as follows: A, control without starter; B, *Aspergillus oryzae* single inoculation group; C, *Bacillus velezensis* and *A. oryzae* individual inoculation group; and D, *B. velezensis* and *A. oryzae* co-inoculation group, during fermentation for 0–2 d.

**Table 1 foods-11-00068-t001:** Changes in moisture, pH, titratable acidity (%), amino-type nitrogen (mg%), and viable microbial counts in *Meju* samples.

Groups	Times (h)	Quality Properties (%, Except for pH)				Viable Microbial Counts (log CFU/g)	
		Moisture	pH	Titratable Acidity	Amino-Type Nitrogen	Aerobic Bacteria	Mold
A	0	60.12 ± 1.90 ^a^	6.52 ± 0.01 ^e^	0.38 ± 0.00 ^ef^	180.0 ± 11.1 ^d^	2.30 ± 0.00 ^bc^	0.00 ± 0.00 ^d^
	24	59.21 ± 2.97 ^a^	6.57 ± 0.00 ^b^	0.42 ± 0.00 ^d^	226.0 ± 11.0 ^bc^	7.93 ± 0.06 ^a^	0.00 ± 0.00 ^d^
B	0	60.03 ± 2.31 ^a^	6.62 ± 0.01 ^a^	0.32 ± 0.01 ^g^	260.0 ± 11.1 ^b^	0.00 ± 0.00 ^c^	5.46 ± 0.17 ^b^
	24	59.96 ± 1.66 ^a^	6.28 ± 0.00^h^	0.78 ± 0.00 ^b^	776.0 ± 11.1 ^a^	8.41 ± 0.26 ^a^	6.45 ± 0.16 ^a^
C	0	58.18 ± 0.71 ^a^	6.54 ± 0.00 ^d^	0.37 ± 0.00 ^f^	227.0 ± 11.1 ^bc^	2.15 ± 3.04 ^bc^	4.67 ± 0.02 ^c^
	24	58.15 ± 1.35 ^a^	6.32 ± 0.00 ^g^	0.95 ± 0.00 ^a^	776.0 ± 11.1 ^a^	9.40 ± 0.02 ^a^	6.44 ± 0.04 ^a^
D	0	58.66 ± 0.32 ^a^	6.46 ± 0.00 ^f^	0.40 ± 0.01 ^e^	196.0 ± 11.1 ^cd^	5.77 ± 0.38 ^ab^	5.62 ± 0.09 ^b^
	24	57.66 ± 0.29 ^a^	6.55 ± 0.00 ^c^	0.68 ± 0.01 ^c^	172.0 ± 0.0 ^d^	9.82 ± 0.05 ^a^	4.31 ± 0.08 ^c^

The experimental groups were divided into four *Meju*: control without a starter (A), *A. oryzae* KCTC 46471 single inoculation Group (B), *B. velezensis* and *A. oryzae* individual inoculation Group (C), and *B. velezensis* and *A. oryzae* co-inoculation Group (D) between pre- and post-fermentation for 0–24 h. Values are presented as means ± standard deviation (number of repetitions = 3). Superscripts in the same column not sharing a common superscript are significantly different at *p* < 0.05 by Duncan’s multiple range test.

## Data Availability

Data is contained within the article.

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
