# Peer review of "Comparative Evaluation of Quality and Metabolite Profiles in Meju Using Starter Cultures of Bacillus velezensis and Aspergillus oryzae"

_foods, 2021, doi:10.3390/foods11010068_

Round 1

Reviewer 1 Report

Line 79: please specify how and where the incubation was performed. Precise the number of days, why 1-2? Was it mixed during this time?

Lines 123-124: If you are looking to analyze volatiles, why add water and NaCl and not directly measure the volatiles on the sample?

Lines 136-137: please give more details about the extraction. How much methanol was added? And how the extraction was done?

Lines 139-143: what kind of metabolites you are looking for? Why not perform HPLC instead?

Line 170: XlStat instead of XlLStat

In table 1 results are given at 0 h and 24 h while the incubation was done for 1-2 days. Please precise.

Figure 2: add Y-axis label

Author Response

Reviewer #1

Comments and Suggestions for Author

Response: We would like to thank the reviewer for the comments and suggestions made about this manuscript. We revised the manuscript according to the suggestion. In order to address the comments clearly, we illustrated the answers for the reviewer’s comments in detail in the manuscript.

Some suggestions for improvement:

Re: According to the suggestion, we made corrections as follows:

  1. Q) Line 79: please specify how and where the incubation was performed. Precise the number of days, why 1-2? Was it mixed during this time?

Re: In Line 79, “for 1 day in fermentation room.”

  1. Q) Lines 123-124: If you are looking to analyze volatiles, why add water and NaCl and not directly measure the volatiles on the sample?

Re: As you already know the principle of SPME is partition (absorption or adsorption) of analytes between the matrix and the SPME fiber coating. So regarding your questions:
1. When one does SPME of volatiles (dissolved in water) it is convenient and smart to add NaCl or other salt to the sample matrix in order to enhance the "escaping" tendency of the volatiles from the matrix (salting out effect) and hence to enhance the partition in to the SPME fiber (or to acheve higher enrichment factors). Therefore, adding salt to the water solution would "push out" the analytes in to the headspace (if you're doing headspace (HS) SPME - very convenient SPME procedure for volatile analysis) or onto the fiber coating itself).

  1. Q) Lines 136-137: please give more details about the extraction. How much methanol was added? And how the extraction was done?

Re: Lines 136-137, Meju at a concentration of 0.02 g/500 μL in 80% methanol.

  1. Q) Lines 139-143: what kind of metabolites you are looking for? Why not perform HPLC instead?

Re: I know that the N-hydroxymethyl derivatives of aldehydes were utilized and the detection limitation of GC was better than that of HPLC.

  1. Q) Line 170: XlStat instead of XlLStat

Re: Correction

  1. Q) In table 1 results are given at 0 h and 24 h while the incubation was done for 1-2 days. Please precise.

Re: change to 24 hours

  1. Q) Figure 2: add Y-axis label

Re: In Fig. 3, we corrected.

Reviewer 2 Report

This article compares inoculation techniques on quality parameters of Meju. 2 organisms are inoculated, Aspergillus oryzae as a monoculture or together with Bacillus velezensis as either direct or co-innoculations. Basic quality parameters of the Meju are measured, ie moisture, acidity, nitrogen as well as volatile profiling as well as growth and identification of microbes present after 24 hours of fermentation.

The paper is well written and methodology sound, however more background is required to describe what the ideal, as in most desirable parameters are for making quality Meju. This should include a brief description of what the current industrial practices are for its production and what technical issues need addressing. Furthermore the authors need to then relate each of their findings to changes to quality parameters in regard to these desirable parameters and thus then provide technical advice in regards to use of these organisms. This is vital to enable a better understanding of the contribution of this work to the field.

Also, please address:

Why is the control (without inoculation) not sterile? What normally happens in industry? What are the indigenous microbes normally present?

Please find further specific comments pertaining to this below.

Please address the below comments to specific sections.

Line:

36: is Meju intentionally underlined?

38: explain what is "prepared koji"

74: expand the explanation of the different inoculation protocols, it is not clear how they differ

79: 1-2 days, please be more specific. Most measurements seem to be made after 24 hours. I would expect he difference in measures at 24 and 48 hours to be quite different

102: change to “…typical colonies identified on each agar plate, taking into account dilutions”

102: Please expand on how you defined “typical colonies”

137: space before °C

156: Move this section to 2.4

183: change “on day 1” to “after 24 hours”

Table 1:

Make superscript ABCD abcd (so as not to confuse with Groups A-D)

218: Why is it improved, please explain what is ideal

233: Please reword “before fermentation was not detected at 24h due to starter control”

248-252. These conclusions don’t make sense to me since abundance of Aspergillus is highest in Condition C where the 2 organisms are individually inoculated. Please explain.

278-287: Please express in a present tense since I assume you are describing how these volatiles are already known to impact Meju (and please reference these). For example, change to “…1 hexanol) have sweetish… and “produce odors” (line 284, add reference), “express a mushroom odor”, “express with fresh grass..”

298: Group B when inoculated with only…

299: Group C inoculated individually…

300: Group D co-inoculated with…

312: “were quickly fermented” What is expected for the duration? How do you define completion? Perhaps write “had significant changes in this time”

333-334: change “levels” to “concentrations”

336: explain significance

388: Reword this sentence

Author Response

Reviewer #2

Comments and Suggestions for Author

Response: We would like to thank the reviewer for the comments and suggestions made about this manuscript. We revised the manuscript according to the suggestion. In order to address the comments clearly, we illustrated the answers for the reviewer’s comments in detail in the manuscript.

Some suggestions for improvement:

Re: According to the suggestion, we made corrections as follows:

  1. Q) 36: is Meju intentionally underlined?

Re: We corrected in article.

  1. Q) 38: explain what is "prepared koji"

Re: We corrected in article.

  1. Q) 74: expand the explanation of the different inoculation protocols, it is not clear how they differ

(C) Each of starter (A. orzyae and B. velezensis) was inoculated into Meju.

(D) Two starters were co-inoculated into Meju at the same time.

  1. Q) 79: 1-2 days, please be more specific. Most measurements seem to be made after 24 hours. I would expect he difference in measures at 24 and 48 hours to be quite different

Re: We corrected in article.

  1. Q) 102: change to “…typical colonies identified on each agar plate, taking into account dilutions”

Re: We corrected in article.

  1. Q) 102: Please expand on how you defined “typical colonies”

Re: We corrected in article.

  1. Q) 137: space before °C

Re: We corrected in article.

  1. Q) 156: Move this section to 2.4

Re: We corrected in article.

  1. Q) 183: change “on day 1” to “after 24 hours”

Re: We corrected in article.

  1. Q) Table 1: Make superscript ABCD abcd (so as not to confuse with Groups A-D)

Re: We corrected in article.

  1. Q) 218: Why is it improved, please explain what is ideal

Re: We corrected in article.

  1. Q) 233: Please reword “before fermentation was not detected at 24h due to starter control”

Re: We corrected in article.

  1. Q) 248-252. These conclusions don’t make sense to me since abundance of Aspergillus is highest in Condition C where the 2 organisms are individually inoculated. Please explain.

Re: We corrected in article.

  1. Q) 278-287: Please express in a present tense since I assume you are describing how these volatiles are already known to impact Meju (and please reference these). For example, change to “…1 hexanol) have sweetish… and “produce odors” (line 284, add reference), “express a mushroom odor”, “express with fresh grass..”

Re: We corrected in article.

  1. Q) 298: Group B when inoculated with only…

Re: We corrected in article.

  1. Q) 299: Group C inoculated individually…

Re: We corrected in article.

  1. Q) 300: Group D co-inoculated with…

Re: We corrected in article.

  1. Q) 312: “were quickly fermented” What is expected for the duration? How do you define completion? Perhaps write “had significant changes in this time”

Re: We corrected in article.

  1. Q) 333-334: change “levels” to “concentrations”

Re: We corrected in article.

  1. Q) 336: explain significance

Re: We added in article.

  1. Q) 388: Reword this sentence

Re: We corrected in article.

Round 2

Reviewer 2 Report

Please thank the authors for addressing some comments, however these still remain:

The introduction should include a brief description of what the current industrial practices are for its production and what technical issues need addressing. Furthermore the authors need to then relate each of their findings to changes to quality parameters in regard to these desirable parameters and thus then provide technical advice in regards to use of these organisms. This is vital to enable understanding of the contribution of this work to the field.

Why is the control (without inoculation) not sterile? What normally happens in industry? What are the indigenous microbes normally present?

71: it is not clear how inoculation of C differs from D

79: "in fermentation room" is this a necessary to state?

102: Please expand in text on how you defined “typical colonies” in comparison to known standards perhaps?

219: Please continue to expand on what the ideal condition is, it is still not clear

233: I still find this section confusing, please reword

281: (ref)

284: are sweet

288: which have fresh grass...

289: Seo et al. Year?

314: Please explain what the time reduction of fermentation was and what the importance of this is.

337: change levels to concentrations

337-339: Explain the relevance/importance of this result

Author Response

I would you like to thank you for your kind comments.
